# The Aerodynamic Gradient Method: Implications of Non-Simultaneous Measurements at Alternating Heights

**Jesper Nørlem Kamp** [1] , **Christoph Häni** [2] , **Tavs Nyord** [1], **Anders Feilberg** [1,*] and **Lise Lotte Sørensen** [3]

1 Air Quality Engineering, Department of Engineering, Aarhus University, 8200 Aarhus N, Denmark; jk@eng.au.dk (J.N.K.); tavs.nyord@eng.au.dk (T.N.)
2 School of Agricultural, Forest and Food Sciences HAFL, Bern University of Applied Sciences, 3052 Zollikofen, Switzerland; christoph.haeni@bfh.ch
3 Department of Environmental Science, Aarhus University, 4000 Roskilde, Denmark; lls@bios.au.dk
* Correspondence: af@eng.au.dk

**Abstract:** Flux measurements with the aerodynamic gradient method (AGM) performed with a single analyzer measuring non-simultaneously at two heights have routinely been conducted. This study investigates the effect of this practice with calculations of single analyzer derived fluxes compared to fluxes derived from simultaneous concentration measurements at two heights for $NH_3$. The results show a mean relative difference of less than 7% for the half-hour averaging intervals, whereas the relative difference in the cumulative loss of total ammoniacal nitrogen (TAN) is less than 4%. Scatter plots and linear regression show linear behavior with slope and intercept close to one and zero, respectively. The regression coefficients were between 0.913 and 0.966 for the simulations, but with large deviations for the single half-hour measurement interval. Changes in the starting height and averaging duration at each height for the single analyzer calculations yield small differences, but the effect is minimal compared to the general uncertainty of flux determination with AGM.

**Keywords:** Ammonia; $NH_3$; ammonia flux; gradient flux; AGM; cavity ring-down spectroscopy

## 1. Introduction

The aerodynamic gradient method (AGM) is a micrometeorological method based on the vertical concentration gradient and the turbulent diffusivity. The AGM method is widely used to determine vertical fluxes of gases over a surface [1–3]. The eddy covariance (EC) method is the direct micrometeorological method that is most commonly used for trace gas fluxes, but fast analyzers (>5 Hz) are required for EC, whereas AGM can use concentration measurements at several heights at lower time resolution. Fluxes derived from EC and AGM measurements represent an average flux of the upwind area from the measurement point and the extent of this area (the flux footprint area) depends on atmospheric conditions, surface characteristics, and measurement height. Thus, horizontal homogeneity, stationarity, and no flux divergence are assumptions for these methods. Furthermore, it implies that the area of interest must have a certain size, with a constant, homogeneous surface flux to fulfill these assumptions. The possibility of lower measurement frequency for AGM offers some advantages in the flux estimation, allowing for flux determination of more compounds or for the usage of low-cost samplers with lower instrument detection limits. Dual instrumentation is needed to measure concentrations at two heights simultaneously. Several studies use a single analyzer to measure concentrations sequentially at minimum two heights as inputs to the AGM [4–6], which provide discontinuous and non-simultaneous concentration measurements at the different heights.

Nelson et al. [4] measured $NH_3$ flux using a cavity ring-down spectrometer (CRDS) with switching heights, resulting in measurements only 40% of the time during each half-hour measurement interval. Kruit et al. [5] measured $NH_3$ flux using denuders in sequential sampling with a cycle time of 10 min for three denuders. Another approach is to use a large buffer volume as Griffis et al. [7] that measured $NH_3$ with CRDS in a sampling sequence with 2 min sampling from each height. Similarly, Zhao et al. [8] used a buffer volume and measured non-sticky compounds ($CH_4$, $CO_2$, and $H_2O$) alternating between two heights in sample cycles of 2 min. Valves were used to change measurement height in flux estimations of $NH_3$, $N_2O$, and $CO_2$ by FTIR spectroscopy in a study by Griffith and Galle [9] in which the fluxes are measured pseudo-simultaneously in a cycle of 20 min for three heights. In a study by Bai et al. [10], $N_2O$ was measured with an open path FTIR, using a motorized mounting head to sequentially measure at two heights. Tanimoto et al. [6] report gradient flux of dimethyl sulfide (DMS) and acetone derived from proton-transfer-reaction mass spectrometry (PTR-MS) sequentially measured at seven heights, and also Omori et al. [11] measured DMS using PTR-MS in sequence with switching valves. In addition to sequential sampling methods, the AGM has been used with automatic denuder systems for continuous measurements [3,12–14]. However, as pointed out by Milford et al. [12], the key challenge in $NH_3$ flux measurements lies in reliable concentration measurements. In a study by Wolff et al. [15] the error of AGM flux measurements introduced by instrument precision and micrometeorological exchange parameters is estimated, and median flux errors of 39% and 50% for $NH_3$ over a grassland and forest site, respectively, are reported. The estimated flux error was between 31% and 68% and the results are comparable to errors reported in similar studies.

Previous $NH_3$ flux measurements have been conducted with the AGM using a single analyzer to measure concentration at minimum two alternating heights resulting in non-simultaneous concentration measurements. In the present paper, we evaluate the effects of non-simultaneous concentration measurements using two CRDS analyzers measuring $NH_3$ concentration simultaneously at two heights over a grass field after slurry application. It is important to look into the implications of this way to use the AGM as a great number of studies have conducted experiments with this practice.

## 2. Measurement Campaigns

Turbulent $NH_3$ fluxes were derived from concentration measurements over the same grass field on two occasions. The 26 ha field is located in central Jutland, Denmark (latitude 56°27′12′′ N, longitude 9°32′26′′ E, elevation 63 m a.s.l.). Two separate measurement campaigns were conducted directly after slurry application in May and August 2019. The experiments with analysis and interpretation of the fluxes have been described in detail by Kamp et al. [16] (under review, *Agric. For. Meteorol.*). Fermented biogas slurry was applied by injection three days after the grass was cut. The field tower with flux instrumentation was placed approximately 200 m from the nearest obstacle in any direction in both measurement campaigns. Wind components were measured at 2 m height at 10 Hz with an ultrasonic anemometer (METEK, uSonic-3 Scientific). Two CRDS analyzers (model G2103, Picarro Inc., Santa Clara, CA, USA) measured $NH_3$ and $H_2O$ concentrations at 1 m and 2 m. Two insulated 10 m PTFE tubes heated to approximately 40°C were used for the inlets. The CRDS analyzers measured side by side at the field before the campaign for comparison and correction of the instruments, see [16] (under review, *Agric. For. Meteorol.*). The performance of this specific CRDS instrument in an agricultural setting has been described in Kamp et al. [17]. The analyzers were placed inside a trailer east of the tower, the least prevailing wind direction.

## 3. The Aerodynamic Gradient Method

The AGM is used to estimate fluxes based on Fick's law and the Monin–Obukhov similarity theory assuming a constant flux layer with the expression

$$F = -K_c \frac{\partial c}{\partial z} \cong \frac{u_* \, k \, (c_2 - c_1)}{\ln\left(\frac{z_2}{z_1}\right) - \psi_{c,2} + \psi_{c,1}} \tag{1}$$

where $F$ is the $NH_3$ flux (µg m$^{-2}$ s$^{-1}$), $K_c$ is the eddy diffusivity (m$^2$ s$^{-1}$), $\partial c/\partial z$ is the concentration gradient of $NH_3$ (µg m$^{-3}$ m$^{-1}$), $u_*$ is the friction velocity (m s$^{-1}$), $k$ is the unit less von Karman constant (0.4), $z_2$ and $z_1$ are upper and lower inlet height, respectively, (m), $c_2$ and $c_1$ are the $NH_3$ concentration measured at heights $z_2$ and $z_1$ (µg m$^{-3}$), and $\psi_{c,2}$ and $\psi_{c,1}$ are the stability correction functions for scalars at heights $z_2$ and $z_1$ [18,19]. The stability correction functions that correct the deviation of the actual vertical concentration profile shape from a neutral profile shape were adapted from Dyer and Hicks [20]. The friction velocity and the Obukhov length are used in the stability correction functions and calculated based on high-frequency wind data. Positive fluxes represent emissions and negative fluxes represent depositions.

## 4. Simulation of Discontinuous AGM

Simultaneous concentration data for $NH_3$ is available at two heights and this dataset is used to calculate AGM flux measurement as if single analyzers switched between two heights. In the study by Nelson et al. [4], a single analyzer measured for 8 min at one height before switching to the second height and measuring at 7 min, then switching back to measure for 7 min and finally measuring for 8 min for each half-hour averaging interval. However, due to response time, only the last 3 min of each interval is used, which gives 12 min of data in each half-hour averaging interval.

Our approach is to simulate measurements of 7.5 or 5 min at each height before switching and only use the last 2.5 min data, giving 10 or 15 min of data in each half-hour averaging interval. We also change the starting position between the upper height (UH) and lower height (LH). In the following, 7.5 min refers to 7.5-minute intervals, where only the last 2.5 min of data is used before switching to the other height, and 5 min refers to 5 min interval, where only the last 2.5 min of data are used before changing to the other height. The flux of $NH_3$ is determined for half-hour averaging intervals.

## 5. Results and Discussion

The emission of total ammoniacal nitrogen (TAN) is estimated as the percentage of the total amount of TAN in the applied slurry. Table 1 shows the loss of TAN in percentage of applied TAN, and Table 2 shows the relative difference of the TAN loss from the simulated fluxes compared to the fluxes derived from AGM. The relative difference in loss of TAN is less than 4% for the different simulations, and the time shift causes a minor difference for the estimates. The best estimates of the TAN loss are produced with 7.5 min interval length in May and 5 min interval length in August. There are 257 and 320 half-hour averaging intervals for May and August, respectively.

**Table 1.** Cumulative loss of total ammoniacal nitrogen (TAN) from field-applied slurry in percentage (%) of total applied TAN for AGM with simultaneous and continuous measurements at two height and simulated discontinuous measurements. UH and LH refer to starting simulation in upper or lower height, respectively.

|  | AGM | 7.5 min UH | 7.5 min LH | 5 min UH | 5 min LH |
|---|---|---|---|---|---|
| May | 8.7 | 8.7 | 8.7 | 9.1 | 8.5 |
| August | 13.1 | 12.9 | 13.2 | 13.1 | 13.2 |

**Table 2.** Relative difference in percentage (%) of the loss of TAN for the simulated fluxes compared to AGM flux. UH and LH refer to starting simulation in upper or lower height.

|  | 7.5 min UH | 7.5 min LH | 5 min UH | 5 min LH |
|---|---|---|---|---|
| May | −0.20 | −0.12 | 3.62 | −3.26 |
| August | −2.13 | 0.50 | −0.08 | 0.37 |

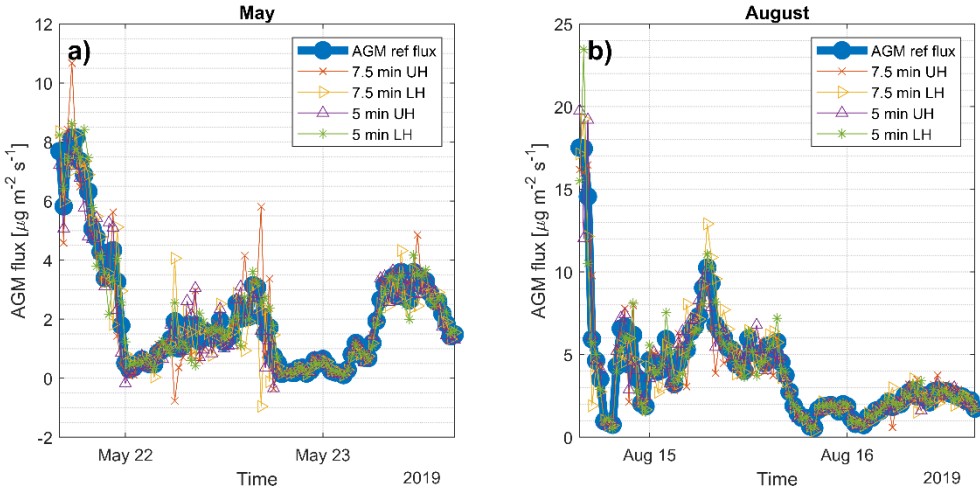

**Figure 1.** Time series of the reference and simulated AGM fluxes for 48 h in (**a**) May and (**b**) August. The thicker blue line is the reference standard AGM flux.

The changes of the simulated and the reference AGM flux over time are shown in Figure 1, where the first 48 hours of data is plotted. Deviations from the reference method is most pronounced for 7.5 min UH and 7.5 min LH that falls in opposite direction of each other. The simulated fluxes are compared to the reference AGM flux by linear regression, where a deviation from a 1:1 line is interpreted as a systematic deviation of the simulated flux to the reference method, and scatter is seen as a random error of the simulated flux. The reference AGM flux is the standard AGM method with continuous measurements at two heights evaluated in half-hour intervals. The mean relative difference in flux is shown in Table 3, where the difference is less than 7% and the UH and LH stating position means spans up to 11% (−7% to 4%). Some discrepancies with the change in starting position and bias in the average flux based on asynchronous concentration measurements are expected due to the limited sample size of the data. Furthermore, rapid increase or decrease in concentration is most likely affecting both measurement heights, but mainly or solely captured at one height due to the discontinuous measurements, thus the changes in the flux is on the same time scale as the switching time. This effect is also visible for the flux estimation in the individual half-hour intervals, where some simulated fluxes deviate largely from the reference method (Figure 2). The impact on the starting position is also an indicator of the uncertainty of flux measurements with a single analyzer. The simulated fluxes are plotted as a function of the AGM flux (see Figure 2). The linear regressions in Table 4 reveal regression coefficients between 0.913 and 0.966. The fluxes used in the statistics in Table 4 are dependent on atmospheric conditions as stability, temperature, precipitation etc. All fluxes are compared for completeness, but conditions favoring removal of $NH_3$ on the same timescale as the switching time are very different from periods with constant flux; thus, this will cause variation in the regression. The uncertainty of the regression is higher after 48 hours compared to the first 48 hours, where the fluxes are larger, see Table 4.

**Table 3.** Mean of relative difference in flux ((simulated−AGM)/AGM) in percentage. UH and LH refer to starting simulation in upper or lower height.

|         | 7.5 min UH | 7.5 min LH | 5 min UH | 5 min LH |
|---------|------------|------------|----------|----------|
| May     | −6.78      | 3.91       | 0.41     | −5.01    |
| August  | −3.61      | 4.14       | −3.11    | 2.73     |

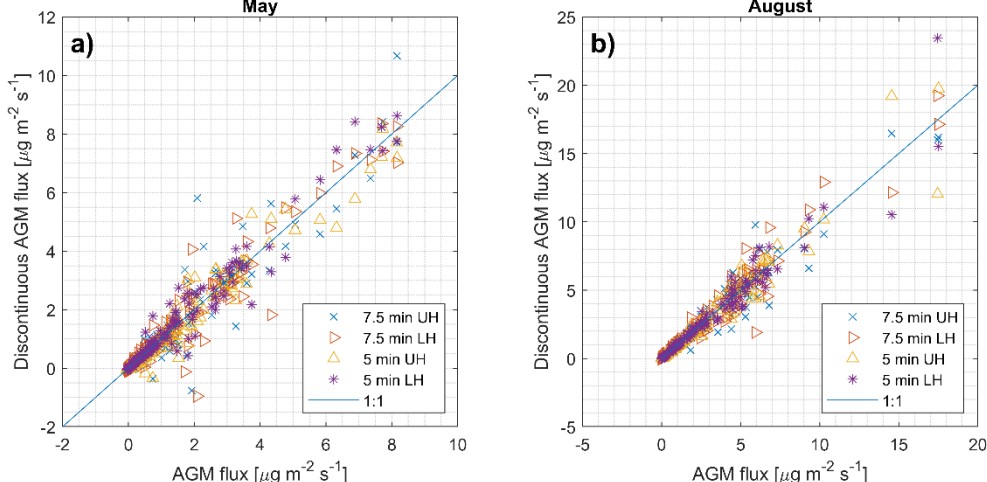

**Figure 2.** Scatter plot of simulated fluxes as a function of the AGM flux in (**a**) May and (**b**) August. The blue line represents 1:1 vertical line. Regression variables are shown in Table 4.

**Table 4.** Linear regression data from the cross-correlation plots, Figure 2, where the simulated flux is a function of the AGM flux. $R^2$ is the coefficient of determination.

| | | All Data | | | First 48 h | | | After 48 h | | |
|---|---|---|---|---|---|---|---|---|---|---|
| | Interval | Slope | Intercept | $R^2$ | Slope | Intercept | $R^2$ | Slope | Intercept | $R^2$ |
| M A Y | 7.5 min UH | 1.02 ± 0.02 | −0.01 ± 0.04 | 0.91 | 0.98 ± 0.02 | 0.00 ± 0.01 | 0.93 | 1.03 ± 0.04 | −0.01 ± 0.1 | 0.87 |
| | 7.5 min LH | 0.99 ± 0.02 | 0.00 ± 0.03 | 0.93 | 1.02 ± 0.02 | −0.01 ± 0.01 | 0.95 | 0.99 ± 0.02 | 0.00 ± 0.03 | 0.90 |
| | 5 min UH | 0.95 ± 0.01 | 0.01 ± 0.02 | 0.96 | 0.98 ± 0.01 | 0.00 ± 0.01 | 0.97 | 0.95 ± 0.03 | 0.02 ± 0.07 | 0.94 |
| | 5 min LH | 1.03 ± 0.01 | 0.00 ± 0.02 | 0.96 | 1.01 ± 0.01 | 0.00 ± 0.01 | 0.97 | 1.02 ± 0.03 | 0.03 ± 0.08 | 0.94 |
| A U G | 7.5 min UH | 0.96 ± 0.01 | 0.03 ± 0.03 | 0.95 | 0.97 ± 0.01 | 0.00 ± 0.00 | 0.98 | 0.94 ± 0.03 | 0.01 ± 0.15 | 0.91 |
| | 7.5 min LH | 1.01 ± 0.01 | −0.01 ± 0.03 | 0.96 | 1.03 ± 0.01 | 0.00 ± 0.00 | 0.98 | 1.03 ± 0.03 | −0.09 ± 0.16 | 0.92 |
| | 5 min UH | 1.00 ± 0.01 | −0.00 ± 0.03 | 0.95 | 0.99 ± 0.01 | 0.00 ± 0.00 | 0.99 | 1.00 ± 0.03 | −0.01 ± 0.16 | 0.91 |
| | 5 min LH | 1.01 ± 0.01 | −0.01 ± 0.03 | 0.96 | 1.01 ± 0.01 | 0.00 ± 0.00 | 0.99 | 1.02 ± 0.03 | −0.05 ± 0.15 | 0.92 |

The use of dual analyzers measurements can potentially be biased due to calibration discrepancies between the instruments; hence, it is a potential advantage to use a single analyzer for discontinuous measurements.

The single analyzer simulations for AGM measurements agreed well with the AGM flux estimations from two continuously running analyzers when comparing the cumulative loss of TAN (Tables 1 and 2) and the mean values of the fluxes (Table 3). However, it should be noticed that for individual half-hour estimates, the relative differences are extremely high for fluxes close to zero, and for fluxes above 0.1 $\mu g\ m^{-2}\ s^{-1}$ the difference is up to 185% in May and 68% in August. Thus, the effect of the deviations from the reference fluxes in the half-hour intervals is minimized over time as the mean flux estimates are within 7%, but for measurements with a small sample size flux estimation with AGM from a single analyzer could introduce a large error. Overall, the error is much smaller than reported for the AGM method itself [15].

The discontinuous measurement with a single analyzer strongly affects individual half-hour flux estimations; however, for measurements over a longer time, where the mean or cumulative flux is the purpose of the measurements only a small error is introduced from measurements at alternating heights with a single analyzer. The uncertainty is much smaller than the uncertainty on the flux measurements itself.

## 6. Conclusions

We conclude that only minor errors are introduced to the flux measurements when AGM is conducted with a single analyzer measuring non-simultaneously at two alternating heights. The mean relative difference in the flux is less than 7% compared to the AGM based on continuous measurements at two heights. Furthermore, starting height and the tested interval length have minimal effect on the uncertainty of the flux measurements.

**Author Contributions:** Conceptualization, J.N.K., C.H. and L.L.S.; methodology, J.N.K.; C.H. and L.L.S.; validation, J.N.K. and C.H.; formal analysis, J.N.K. and C.H.; investigation, J.N.K.; resources, A.F.; T.N.; L.L.S.; data curation, J.N.K.; writing—original draft preparation, J.N.K.; writing—review and editing, J.N.K.; C.H.; T.N.; A.F. and L.L.S.; visualization, J.N.K. and C.H.; supervision, A.F.; T.N.; L.L.S.; project administration, J.N.K. and A.F.; funding acquisition, A.F.; T.N.; All authors have read and agreed to the published version of the manuscript.

**Funding:** This research was funded by Innovation Fund Denmark, grant number 6150-00030A and by GUPD, grant number 34009-16-1112.

**Acknowledgments:** Thanks to technicians Bjarne Jensen, Heidi Grønbæk, and Peter Storegård Nielsen for helping with preparation and the setup of the equipment. Thanks to the local house owners Morten and Esben for supplying us with electricity at the field. Thanks to senior advisor Michael Nørremark for measuring GPS positions on the field.

**Conflicts of Interest:** The authors declare no conflict of interest.

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
