# Peer review of "The Aerodynamic Gradient Method: Implications of Non-Simultaneous Measurements at Alternating Heights"

_atmosphere, doi:10.3390/atmos11101067_

Round 1
Reviewer 1 Report
This is a nice Short Communication that describes results of a comparison of continuous gradient measurements at two levels with simulated intermittent / sequential sampling at the same two levels, for the purpose of calculating aerodynamic gradient fluxes. The latter method is quite commonly applied but infrequently validated, so this manuscript represents a welcome verification.
The manuscript is well written, logically structured, and requires very few modifications to be publishable. A few notes below.
The correct preposition to use with "height" is "at", not "in", e.g. line 37, "at several heights". [several instances]
L46-49: awkward. Break up into 2 or 3 sentences.
L63: [11] duplicated
L86 and elsewhere: the links to the figures and tables don't work; getting "Error! Reference source not found". Type this out.
L142: (7% to 4%)
To finish, a couple of analytical questions/suggestions:
In section 5, at Table 1 or nearby, list the number of half-hours (N) of data available/incorporated for each month. It would be useful (more useful than Fig 1!) to add a time series figure that shows a few days of NH3 fluxes, to provide the reader with a feeling for how constant (or not) the fluxes were. What is not discussed is that the comparison statistics shown here cannot be generalized, since they are very dependent on how steady the fluxes are. If fluxes are invariant, the two methods will compare perfectly; if they vary significantly, especially on a time scale comparable to the switching time, there could be large differences. What could strengthen the paper is a bit of additional analysis along the following lines: subdivide the monthly data record into subsections (e.g. weekly), do the same statistics, and compare. Or pick periods with relatively constant fluxes vs. rapidly changing fluxes, and do the statistics separately for each subset.
Lastly, I was a bit puzzled by the attention given to the starting position. It seems to me that whether you start a month-long time series 7.5 minutes earlier or later (if you only have one instrument to measure both levels) will make little difference. For much shorter analysis periods, the difference may of course be significant. If I missed something, perhaps this aspect should be explained a bit better.
Reviewer 2 Report
Manuscript 951437
Title: The aerodynamic gradient method: implications of non-simultaneous measurements at alternating heights
In this short communication the authors examine the impacts of using non-simultaneous measurements of NH3 on the resulting flux calculations of the aerodynamic gradient method. Overall, the manuscript is concise and well written. This work is beneficial to the community as it summarizes a simple, yet informative, analysis that investigates potential bias in an observational platform that is commonly used the field.
More specific comments for the authors are provided below and mostly consist of clarifications needed to make the material clearer and complete as a standalone publication.
- There are several ‘Reference source’ errors throughout that make the text hard to follow.
- Line 138 – It is not clear from the text what the ‘reference AGM flux’ is. Is this the flux calculated using the standard AGM method and the full 30-minute intervals, or whatever the higher duty cycle is? Lines 161-162 seem to clarify this (although it is hard to say with the reference errors), but this should be made clear when the concept is first introduced.
- There is no discussion about the calibration (or cross calibration) of the two CRDS instruments used in this analysis. Are these details included in paper under review in Agric. For. Meteorol.? One could argue that a potential advantage of using a single instrument is that there would not be calibration discrepancies between the measurements at the two heights. Even a brief statement about the calibration procedure would make the discussion around dual analyzer (simultaneous) and single analyzer (discontinuous) methods more comprehensive.
- Line 168 – The authors state that flux estimations using a discontinuous approach with a small sample size can lead to large errors. As a clarification, what metric of the sampling are you referring? Put another way, does ‘small’ and ‘individual measurement periods’ refer to single day measurements, single half hour interval, etc.? Defining this in the text would help the reader understand when the single analyzer, discontinuous method is not appropriate. As the text stands now, it is not clear what the distinction is between “individual measurement periods” and when large differences are mitigated by averaging “over time.”
